# Evaluation of the Effect of Supervised Group Exercise on Self-Reported Sleep Quality in Pregnant Women with or at High Risk of Depression: A Secondary Analysis of a Randomized Controlled Trial

**DOI:** 10.3390/ijerph19105954

**Published:** 2022-05-13

**Authors:** Lotte Broberg, Peter Damm, Vibe G. Frokjaer, Susanne Rosthøj, Mie Gaarskjaer de Wolff, Stinne Høgh, Ann Tabor, Hanne Kristine Hegaard

**Affiliations:** 1Department of Obstetrics, Copenhagen University Hospital–Rigshospitalet, 2100 Copenhagen, Denmark; pdamm@dadlnet.dk (P.D.); stinne.hoegh@regionh.dk (S.H.); hanne.kristine.hegaard@regionh.dk (H.K.H.); 2The Interdisciplinary Research Unit of Women’s, Children’s and Families’ Health, Copenhagen University Hospital–Rigshospitalet, 2100 Copenhagen, Denmark; 3Center for Clinical Research and Prevention, Frederiksberg Hospital, 2000 Frederiksberg, Denmark; 4Department Clinical Medicine, University of Copenhagen, 2200 Copenhagen, Denmark; ann.tabor@regionh.dk; 5Neurobiology Research Unit, Department of Neurology, Copenhagen University Hospital–Rigshospitalet, 2100 Copenhagen, Denmark; vibe.froekjaer@regionh.dk; 6Mental Health Services, Capital Region of Copenhagen, 2605 Brondby, Denmark; 7Danish Cancer Society Research Center, 2100 Copenhagen, Denmark; suro@cancer.dk; 8Department of Obstetrics and Gynecology, Copenhagen University Hospital–Hvidovre, 2650 Hvidovre, Denmark; mie.gaarskjaer.de.wolff@regionh.dk; 9Center of Fetal Medicine, Department of Obstetrics, Copenhagen University Hospital-Rigshospitalet, 2100 Copenhagen, Denmark

**Keywords:** exercise, sleep quality, depression, self-reported, patient reported outcomes, pregnancy

## Abstract

Poor sleep quality is common during pregnancy. Our objective was to evaluate the effect of supervised group physical exercise on self-reported sleep quality in pregnant women with or at high risk of depression, and secondly, to describe the association between sleep quality and psychological well-being during pregnancy and postpartum. This was a secondary analysis of a randomized controlled trial (*n* = 282) (NCT02833519) at Rigshospitalet, Denmark. Sleep quality was evaluated using the Pittsburgh Sleep Quality Index (PSQI), psychological well-being by the five-item WHO Well-Being Index (WHO-5). The intention-to-treat analysis showed no difference in mean global PSQI score neither at 29–34 weeks, 6.56 (95% CI: 6.05–7.07) in the intervention group and 7.00 (95% CI: 6.47–7.53) in the control group, *p* = 0.2, nor at eight weeks postpartum. Women with WHO-5 ≤ 50 reported higher mean global PSQI scores at baseline, 7.82 (95% CI: 7.26–8.38), than women with WHO-5 score > 50, mean 5.42 (95% CI: 5.02–5.82), *p* < 0.0001. A significant difference was also present post-intervention and eight weeks postpartum. No significant effect of group exercise regarding self-reported sleep quality was seen at 29–34 weeks of gestation or postpartum. Low psychological well-being was associated with poor sleep quality during pregnancy and postpartum.

## 1. Introduction

As a restorative health behavior, sleep is associated with numerous physical and mental health outcomes [1]. Inadequate sleep is a prominent and increasing public health problem, linked to an increased risk of obesity, diabetes, cardiovascular disease, and breast cancer [2,3,4]. The association between poor mental health and poor sleep quality is described as bidirectional, where inadequate sleep can be both a causal contributor to, and a symptom of, psychiatric disorders such as depression and anxiety [5,6]. A recent systematic review stressed the possible association between impaired sleep and self-injury [7]. Women have a two-fold risk of sleep disturbances compared with men [8], and the risk is further exacerbated during pregnancy and postpartum [9].

A 2018 meta-analysis found that 45.7% of pregnant women experienced poor overall sleep quality, defined as a global score ≥ 5 on the Pittsburgh Sleep Quality Index (PSQI) and as a decrease in sleep quality from the second to third trimester [9]. Sleep quality is negatively affected during pregnancy, putatively due to the mechanical, hormonal, vascular, and metabolic changes [10]. Poor sleep quality has been associated with an increased risk of cesarean section [11], prolonged labor [12], and a recent systematic review found a moderate association between poor sleep and perinatal depression [13]. Further, poor sleep postpartum increases the risk of depression and anxiety [14]. Given the potential for prevention and health promotion, as well as recognizing the concerns of pregnant women and healthcare professionals about the use of sleep medication during pregnancy [15,16], there is a need for interventions to improve sleep quality during pregnancy. Though the effect is not fully understood, exercise holds promise as a modifiable behavioral factor promoting sleep, and resistance exercise has been shown to improve sleep quality, to a moderate effect, in the general population, irrespective of sex [17,18]. Among individuals with severe mental illness, a systematic review from 2019 found that exercise had a strong, positive effect on sleep quality [19].

The basic mechanisms underlying the exercise–sleep relationship are not fully understood, but it has been suggested that the gradual decline in body temperature occurring after exercising contributes to drowsiness and facilitates sleep [20]. Another possible mechanism is that exercise improves sleep quality by reducing anxiety [21].

It is recommended to conduct at least 30 min of moderate intensity exercise daily through a normal, uncomplicated pregnancy. However, many pregnant women do not meet these recommendations [22], among other factors due to family obligations and their relatives’ perception of exercise being risky during pregnancy [23]. This indicates that women’s physical activity level during pregnancy needs attention. A study found that specifically endurance and resistance exercise had a strong effect in relation to improving sleep quality during pregnancy [24]. Among pregnant women with depression or a history of mental disorders, the evidence of exercise as a contributor to improved sleep quality is sparse. A randomized controlled trial (RCT) (*n* = 92) found a small effect of yoga in relation to sleep disturbances in pregnant women with depression [25], and larger RCTs and replications are required to determine the clinical potential of exercise as a means to improve sleep quality among pregnant women with or at high risk of depression. From that background, we hypothesized that supervised group exercise twice weekly for twelve weeks would improve self-reported sleep quality in pregnant women with current or a history of depression and/or anxiety. Secondly, we hypothesized that the strength of the sleep–psychological health association would be moderate.

Our objective was to evaluate the effect of a supervised group exercise intervention on self-reported sleep quality among pregnant women with current or a history of depression and/or anxiety. A secondary aim was to describe the association between self-reported sleep quality and psychological well-being in the total study population during pregnancy and postpartum.

## 2. Materials and Methods

### 2.1. Study Design

The present study is a planned secondary analysis of an RCT, the EWE Study. The EWE Study´s primary objective was to evaluate the effect of a supervised group exercise intervention on psychological well-being and symptoms of depression among pregnant women with or at high risk of depression. A detailed study protocol [26] and the primary outcome article have been published [27]. In brief, the EWE Study was a parallel-group RCT conducted from August 2016 to September 2018, with follow-up until April 2019, at Rigshospitalet, Denmark. Rigshospitalet serves as a tertiary referral center and is the primary birth facility of central Copenhagen residents, with 5406 deliveries in 2018.

Eligible participants were pregnant women with current or a history of depression and/or anxiety within the last ten years, requiring treatment by a psychiatrist, a general practitioner, or a psychologist, and/or use of antidepressants three months prior to or during pregnancy. Depression and anxiety were defined according to the *Diagnostic and Statistical Manual of Mental Disorders*, Fifth Edition [28]. Additional criteria for inclusion were: singleton pregnant women, gestational age between 17 and 22 weeks at the start of the intervention, ≥18 years of age, and sufficient Danish language skills. Women who also had a chronic somatic medical condition were only included after consultation with an obstetrician. Women were excluded if they had a history of substance abuse, eating disorders, severe medical or obstetric complications, pelvic girdle syndrome (diagnosed by a physiotherapist or a physician) during the current or a previous pregnancy, known fetal chromosomal anomalies, or fetal malformations. After randomization, participants were excluded from the intervention for the following reasons: pelvic girdle syndrome; preeclampsia; vaginal bleeding; or other symptoms that contraindicated physical activity [26].

A total of 647 pregnant women were invited to participate, of whom 282 women gave written informed consent and completed a baseline questionnaire providing information on self-reported sleep quality, psychological well-being, and maternal characteristics such as age, body mass index (BMI), educational level, and level of physical activity. The main reason for declining to participate was lack of time (Figure 1).

To provide a fair comparison between the intervention and the control group, the distribution of known and unknown prognostic factors was balanced, on average, at baseline by using randomly permuted block randomization (block size four, six, or eight), ensuring proper allocation sequence concealment. The participants were randomly assigned to either supervised group exercise (*n* = 143) or the control group (*n* = 139) (Figure 1). Eight women, all from the intervention group, withdrew consent: one found the exercise uncomfortable to perform; two preferred other kinds of physical activity; and five could not find the time to participate. A total of 270 women were included in the intention-to-treat analyses (Figure 1). In the intervention group, six women were withdrawn from the intervention, but not from the intention-to-treat analysis, because of vaginal bleeding (*n* = 2), threatened preterm labor (*n* = 2), and pelvic girdle syndrome (*n* = 2). For comparison, four women in the control group met the withdrawal criteria: two women with vaginal bleeding; one with hypertension; and one with pelvic girdle syndrome. The four women were included in the intention-to-treat analysis.

### 2.2. Intervention Group

As a supplement to the usual antenatal care for women with current or a history of depression and/or anxiety, the women in the intervention group were offered in-hospital supervised group exercise twice weekly for 12 weeks starting from 17–22 weeks of gestation. Four physiotherapists from Rigshospitalet developed and supervised the exercise intervention, in accordance with the Danish national recommendations for exercise during pregnancy [29]. Each session lasted 70 min and comprised a 10-min warm-up (Borg Rating of Perceived Exertion (RPE) 7–10) [30], 20 min of endurance training on treadmills, exercise bikes, or cross trainers (Borg RPE 11–15), 25 min of strength training (back, abdomen, thighs, arms, and pelvic floor), and 15 min of stretching and relaxation (Borg RPE 6). The women in the intervention group were carefully introduced to the meaning of 60–70% of one repetition maximum (RM) by the physiotherapists and the individual suitable weight for exercises was found for each participant The study protocol describes the specific training exercises and their duration in detail [26]. Participants received a supportive weekly email to improve adherence, and attendance was recorded at each session by the physiotherapists as a standard procedure.

### 2.3. Control Group

Women in the control group were provided with the usual antenatal care for pregnant women with current or a history of depression and/or anxiety based on interdisciplinary collaboration and coordinated by a specialized midwife at the Department of Obstetrics, Rigshospitalet. According to national recommendations, these visits included general individual advice, provided verbally, regarding physical exercise 30 min daily during pregnancy [29].

### 2.4. Outcomes

This secondary analysis from the EWE study reports the predefined outcome, self-reported sleep quality, measured using PSQI to determine sleep quality and sleep disturbances over a period of one month [31]. PSQI, which has good construct validity and reliability for assessing sleep quality among pregnant women [32], contains 19 items assessing a wide variety of factors relating to sleep quality and measures seven individual components of sleep: subjective sleep quality; sleep latency; sleep duration; habitual sleep efficiency; sleep disturbances; use of sleeping medication; and daytime dysfunction. The seven individual component scores are rated from 0–3 (0 indicates no difficulties, 3 indicates severe difficulties) and then summed as a global PSQI score ranging from 0 to 21. This score discriminates good sleep (score ≤ 5) from poor sleep (score > 5), with a sensitivity of 89.6% and a specificity of 86.5% [31]. Based on this, we predefined poor sleep as a cut-off score of >5, before conducting the analysis. Beyond the global PSQI score, component 1 also measure subjective sleep quality, here based on the one single question: “During the past month, how would you rate your sleep quality overall?” with the possible answers (very good, fairly good, very bad, or fairly bad). Component 1 was dichotomized (predefined before conducting the analysis) to distinguish “no sleep problems” (very good/fairly good subjective sleep quality) from “sleep problems” (very bad/fairly bad subjective sleep quality). This dichotomization was based on the clinical relevance of discriminating no sleep problems from sleep problems and was conducted similarly in a previous Danish study [33]. PSQI scores were obtained using an online self-administered questionnaire at baseline, at 29–34 weeks of gestation, and at eight weeks postpartum.

Subjective psychological well-being covering the preceding two weeks was measured by the five item World Health Organization Well-Being Index (WHO-5) [34], which was the primary outcome of the RCT [27]. A WHO-5 score ≤ 50 indicates reduced psychological well-being and is the cut-off score when applying WHO-5 to screen for depression [34].

### 2.5. Statistical Analysis

For incomplete questionnaires, multiple imputation of missing items was performed for each time point separately. Missing items were imputed using fully conditional mean imputation [35]. Fifty complete versions of the questionnaires were generated and sum scores were determined for each of the imputed datasets. Analyses of the sum scores for the imputed datasets were combined using Rubin’s rule [35].

To compare the outcomes at 29–34 weeks of gestation and eight weeks postpartum between the two intervention groups, linear mixed models were applied. To gain efficiency in the analyses and to account for potential baseline imbalances, the baseline scores can be included in the analyses [36]. It has been demonstrated that the optimal way to adjust for baseline variables is to include these in the linear mixed models as outcomes rather than as covariates [37]. Due to the randomization, the means of the baseline scores in the two intervention groups are equal by design. Therefore, the linear mixed model also including the baseline scores as outcomes should be constrained to have equal means at baseline in the two groups. The model for the mean structure included the interaction between groups (intervention and control) and the time (baseline/29–34 weeks of gestation/eight weeks postpartum), with the constraint that the means in the two groups were assumed to be equal at baseline due to randomization. An unstructured covariance pattern was used to model the correlation between the measurements for each participant. For each group and each time point, the proportion of women with a PSQI score > 5 was determined using a logistic regression model with parameters estimated by weighted generalized estimating equations to account for repeated measures and missing data [38]. The weights were defined as the inverse probabilities of being observed conditional on previous measurements of PSQI (quantitative), treatment group, and previous missing value of PSQI and were estimated from logistic regression models. An unstructured correlation matrix was used as the working correlation.

A per protocol analysis was performed comparing mean global PSQI for the subgroup of women attending > 74% of the sessions to the control group. The mean structure included the interaction between time and randomization group. The analysis was performed unadjusted and, based on existing evidence, adjusted for physical activity before pregnancy (yes/no), psychological well-being at baseline (WHO-5 score ≤ 50 vs. WHO-5 score > 50), and educational level (advanced degree and 3–4 years’ higher education vs. 1–2 years’ higher education, skilled worker, and compulsory education).

Using linear respective logistic regression, we compared the mean global PSQI score and the proportion of women with PSQI score > 5 for women with baseline WHO-5-score ≤50 and >50, respectively.

A *p* value of <0.05 was considered statistically significant. Statistical analyses were performed using SPSS version 26 and R version 3.5.2.

## 3. Results

### 3.1. Participant Characteristics

Table 1 presents the comparable baseline maternal characteristics of the intervention group and the control group. At baseline, the study population’s mean global PSQI score was 6.26, the mean WHO-5 score was 55.2, and 78% were physically active ≥ 3.5 h per week before pregnancy (Table 1).

### 3.2. Response Rate, Adherence to Intervention, and Amount of Weekly Exercise

The response rate at 29–34 weeks of gestation was 95% (127/133) in the intervention group and 86% (118/139) in the control group (Figure 1), 20 questionnaires had a single missing item, 17 had 2–4 missing items. At eight weeks postpartum the response rate was 74% (99/133) and 61% (84/137), respectively (Figure 1), 23 questionnaires had a single missing item, 9 had 2–4 missing items. In the intervention group 55 (42%) attended >74% of the exercise sessions, 47 (35%) attended 50–74% of the sessions, and 31 (23%) fewer than half of them. The median weekly amount of physical activity, taking the physical exercise in the intervention program into account, was determined at the end of the intervention period and again at eight weeks postpartum. The weekly median amount of physical activity at 29–34 weeks of gestation was 4 h for both groups but with a range of 0–24 in the intervention group and a range of 0–16 in the control group. At eight weeks postpartum, it was 6 h (range 0–25) for both the intervention group and the control group.

### 3.3. Intention-to-Treat Analysis

At 29–34 weeks of gestation the intention-to-treat analysis showed no difference in mean global PSQI score with 6.56 (95% CI: 6.05–7.07) in the intervention group, and 7.00 (95% CI: 6.47–7.53) in the control group, *p* = 0.2 (Table 2). The prevalence of women with poor sleep (global PSQI score > 5) at 29–34 weeks of gestation was 56.8% (*n* = 71) in the intervention group and 64.1% (*n* = 75) in the control group, *p* = 0.24. No differences were found between the two groups regarding the seven individual PSQI components although subjective sleep quality (component 1, dichotomized) and sleep duration (component 3) tended to be better in the intervention group, *p* = 0.06 (Table 2).

Eight weeks postpartum we found no differences in the mean global PSQI score and poor sleep quality (global PSQI score > 5) between the two groups (Table 2). No significant differences were found eight weeks postpartum between the two groups for the following six components: subjective sleep quality; sleep latency; sleep duration; sleep disturbances; use of sleeping medication; and daytime dysfunction. We found a poorer habitual sleep efficiency (component 4) in the intervention group compared to the control group (Table 2). In both groups, we found that overall sleep quality decreased from baseline to eight weeks postpartum. We found a significantly higher mean global PSQI score in the intervention group eight weeks postpartum, mean 7.69 (95% CI: 7.14–8.25) compared to 29–34 weeks of gestation, mean 6.56 (95% CI: 6.05–7.07), *p* < 0.0001. In the control group, the mean global PSQI score did not differ significantly from 29–34 weeks of gestation to eight weeks postpartum (Figure 2).

### 3.4. Per Protocol Analysis

The pre-specified per protocol analysis including women attending >74% of the exercise sessions showed a significantly lower mean global PSQI score 6.06 (95% CI: 5.27–6.85) compared with the control group 7.04 (95% CI: 6.50–7.57) at 29–34 weeks of gestation, *p* = 0.04. When adjusting for educational level, physical activity before pregnancy, and baseline WHO-5 score, the difference between the groups was attenuated, *p* = 0.06. The proportion of women with poor sleep (global PSQI score > 5) was 58.2% (*n* = 32) in the intervention subgroup and 64.1% (*n* = 75) in the control group, *p* = 0.49. Eight weeks postpartum, the mean global PSQI score in the intervention subgroup was 7.81 (95% CI: 6.97–8.66) compared with the control group mean of 7.60 (95% CI: 6.97–8.23), *p* = 0.69. The proportion of women with poor sleep (global PSQI score > 5) was 76.7% (*n* = 33) in the group attending > 74% of the exercise sessions and 78.3% (*n* = 65) in the control group, *p* = 0.83. No differences were found between groups according to the individual PSQI components eight weeks postpartum in the per protocol analysis.

### 3.5. Association between Subjective Sleep Quality and Psychological Well-Being in the Total Study Population

In the total study population sample, we found that women with a WHO-5 score ≤ 50 [30] reported a significantly higher mean global PSQI score at baseline, 7.82 (95% CI: 7.26–8.38), than women with high psychological well-being (WHO-5 score > 50), mean, 5.42 (95% CI: 5.02–5.82), *p* < 0.0001. A significant difference was also present at 29–34 weeks of gestation, and eight weeks postpartum, *p* < 0.0001. At baseline, the proportion of women with a global PSQI score > 5 differed significantly; 75.2% of women had a WHO-5 score ≤50, and 47.3% had a WHO-5 score > 50, *p* < 0.0001. A significant difference was also present at 29–34 weeks of gestation, *p* = 0.03, while no difference was seen eight weeks postpartum, *p* = 0.24 (Table 3).

## 4. Discussion

At 29–34 weeks of gestation, we found no effect of the 12-week, supervised group exercise on self-reported sleep quality measured as global PSQI mean score and frequency of poor sleep, global PSQI score > 5. Conversely, a 2020 systematic review and meta-analysis with six RCTs (*n* = 688), including one study with clinically depressed pregnant women, found that overall exercise (aerobics, yoga, tai chi, and gymnastics training) among pregnant women contributed to improved sleep quality. However, the authors interpreted their results with caution, reasoning that few high-quality studies were included in their analysis [24]. Our RCT assessed an exercise intervention in a high-risk group of pregnant women, which is relevant in terms of developing a strategy for preventing complications, such as perinatal depression [13]. Although we did not find an effect of the intervention according to global PSQI mean score and poor sleep, we found a trend towards a lower proportion of women who reported their sleep as very bad or fairly bad. Further, we found a tendency towards longer sleep duration in the intervention group compared with the control group at 29–34 weeks of gestation. A study found that longer sleep duration was positively associated with psychological well-being, and, although the study was conducted among non-pregnant women [39], the results might also be applicable to pregnant women. Additionally, our unadjusted pre-specified per protocol analysis of women who attended >74% of the exercise sessions showed a significantly better overall sleep quality measured by the global PSQI mean score compared with the control group at 29–34 weeks of gestation. However, the difference was attenuated and no longer statistically significant after adjustment for the potential confounders, based on the literature. Nevertheless, this may indicate that regular exercise can have a positive effect on sleep quality, as seen in previous research [40]. Further, a small study in a population of non-pregnant women with insomnia showed that sleep influenced exercise the next day, rather than the other way around [41]. It is possible that the women who attended >74% of the exercise sessions had high adherence because of higher sleep quality or better psychological well-being, as seen in the primary outcome article [27], which is why reversed causality cannot be ruled out. We are aware that the per protocol analysis represents a selected subgroup, and we interpret these results with caution [42].

In both groups, we found that overall sleep quality decreased from baseline to eight weeks postpartum. This finding is in line with a recent study measuring sleep quality using PSQI [43] and with a meta-analysis [9], and is due to several factors such as weight change, nocturia, restless leg syndrome, etc. [10]. Notably, we found that the intervention group reported a significantly poorer sleep quality eight weeks postpartum compared with 29–34 weeks of gestation and poorer habitual sleep efficiency (component 4) compared with the control group. In the control group, there was no significant difference in reported sleep quality in the same period. This indicates that the decrease in sleep quality from late pregnancy to eight weeks postpartum in the intervention group and the poorer habitual sleep efficiency (component 4) may be associated with cessation of the intervention. Likewise, a previous study found that a decrease in physical activity in general was associated with poorer sleep [44]. We did not see a decrease in the weekly amount of exercise, but a change in intensity cannot be ruled out. Mothers in Denmark are on average on maternity leave for ten months after giving birth, so we have no reason to believe that returning to work has influenced our results. It cannot be ruled out that mechanical, hormonal, vascular, and metabolic changes occurring during pregnancy and postpartum have influenced the results. However, due to the randomization we assume these factors to be equally distributed in both groups As moderate-intensity exercise postpartum has the potential to improve sleep quality [24], thereby decreasing the risk of perinatal depression [13], it might have been useful to continue the exercise intervention postpartum.

We found a high mean global PSQI in both groups [31], but it was still comparable to previous studies (ranging from 6.1–8.3), even among pregnant women without psychiatric disorders [45,46]. However, we found a larger proportion of women with a global PSQI score > 5 in both groups at 29–34 weeks of gestation (56.8% and 64.1%) compared with the 45.7% found in a meta-analysis of pregnant women, primarily without signs of depression and where a PSQI score of >5 was used (the ≥5 cut off in the article by Sedov et al. was a typographical error which was clarified after personal communication with the author) [9]. Our finding of a high proportion of women with poor sleep, defined as a global PSQI score > 5, may be due to our study population having current or a history of depression and/or anxiety [47]. However, in this and other studies [9,46], the findings of an average PSQI score above the cut-off score, used to differentiate between good and poor sleepers, may reflect that the PSQI is a generic tool that does not take all the changes, e.g., hormonal and metabolic, that occur during pregnancy and postpartum into account. Although the high mean PSQI score may appropriately reflect that poor sleep is common during pregnancy, it complicates the task of determining which expectant mothers require further assessment and treatment. As expected, we found that women with reduced psychological well-being (WHO-5 score ≤ 50) [34] at baseline reported a significantly higher mean global PSQI score than women with high psychological well-being (WHO-5 score > 50). This is in line with previous studies finding an association between low psychological well-being and poor sleep [48,49], however, we found a larger proportion of women and a stronger association than expected. A significant difference was also present at 29–34 weeks of gestation, and eight weeks postpartum. A study found that women with low psychological well-being eight weeks postpartum were six times more likely to report that their newborns’ sleep patterns did not allow them to sleep well [50]. However, the direction in the association between low psychological well-being and maternal experience of poor sleep was not clear [50].

### 4.1. Strengths and Limitations

#### 4.1.1. Strengths

One of the strengths of this study is that it was a pre-specified secondary analysis based on a large, well-conducted RCT. It is the largest RCT to date measuring PSQI among pregnant women with, or at high risk of, depression, however, power calculation was not made for the secondary outcome PSQI. We consider it a strength, that PSQI is measured twice during pregnancy and again postpartum, which provides insight into the changes in sleep quality during this period. Further, it is a strength, that the physiotherapists registered participation in the exercise sessions, and thereby enhanced data accuracy and intervention fidelity. According to the statistical analysis, it is a strength that we dealt with missing PSQI items and missing questionnaires, and thereby reduced the risk of bias [51].

#### 4.1.2. Limitations

One of the limitations of the study is that the PSQI is a retrospective measurement and only measures self-reported subjective sleep quality. However, using an objective measurement of sleep, such as actigraphy, would be difficult with a present sample size such as ours. Further, studies found that subjective sleep quality was more strongly associated with depression postpartum than objective sleep measures [52]. The PSQI is a validated patient-reported outcome measure widely used as an outcome in the obstetric field [32], however, during pregnancy, the related comorbid conditions of being pregnant are likely to influence the results. The study population was well-educated, had normal BMI, was largely physically active before, during, and after pregnancy, was primarily primiparous, and was proficient in Danish, limiting the generalizability of the results to other populations. A large proportion of eligible women declined to participate and, unfortunately, the Ethics Committee of the capital region of Denmark did not permit us to collect baseline characteristics of the invited women choosing not to participate. Only 25% of the study population was multiparous, even though approximately 40% of women referred to Rigshospitalet are multiparous [22], indicating that primarily nulliparous women agreed to participate. Time factors are known reasons for not participating in a research project [53], which might explain why more multiparous than nulliparous women did not accept the invitation. It cannot be ruled out that this preponderance of primipara in the study population may have entailed sample selection bias and potentially affected the results. As the control group reported the same weekly amount of physical activity as the intervention group, a Hawthorne effect cannot be ruled out [54]. Participation in the study itself might have provided motivation for some of the women to increase their exercise level, however, to reduce this risk we did not introduce the control group to the exercise program.

## 5. Conclusions

Supervised group exercise did not improve self-reported sleep quality, either at 29–34 weeks of gestation or at eight weeks postpartum among pregnant women with or at high risk of depression. We found that overall sleep quality decreased from baseline to eight weeks postpartum and, as expected, that poor sleep was particularly pronounced for the pregnant women with low psychological well-being. Our data do not support the introduction of physical exercise into clinical practice in order to improve sleep quality among pregnant women with or at high risk of depression.

Sleep complaints are common during pregnancy, but it needs attention in antenatal care that sleep quality further decreases postpartum and that poor sleep quality is particularly pronounced for women with depression or low psychological well-being. While poor sleep is both a symptom of, and a causal contributor to, depression, health care professionals should give attention to the risk of a vicious circle. Further studies are needed to examine the possible effect of exercise on sleep quality.

## Figures and Tables

**Figure 1 ijerph-19-05954-f001:**
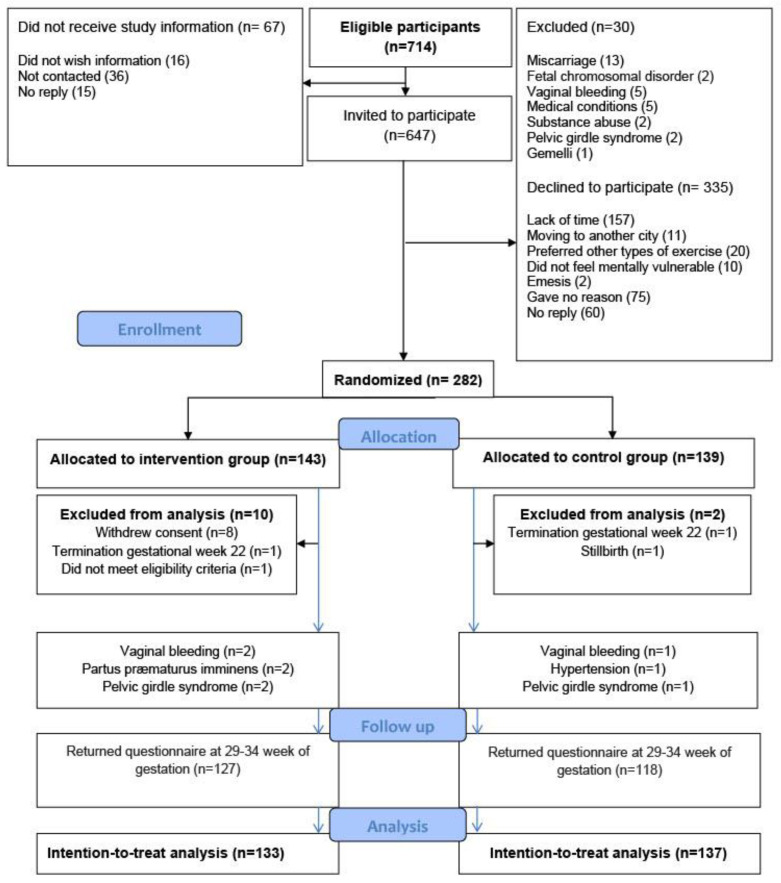
Flow chart for the randomized controlled trial, the EWE Study.

**Figure 2 ijerph-19-05954-f002:**
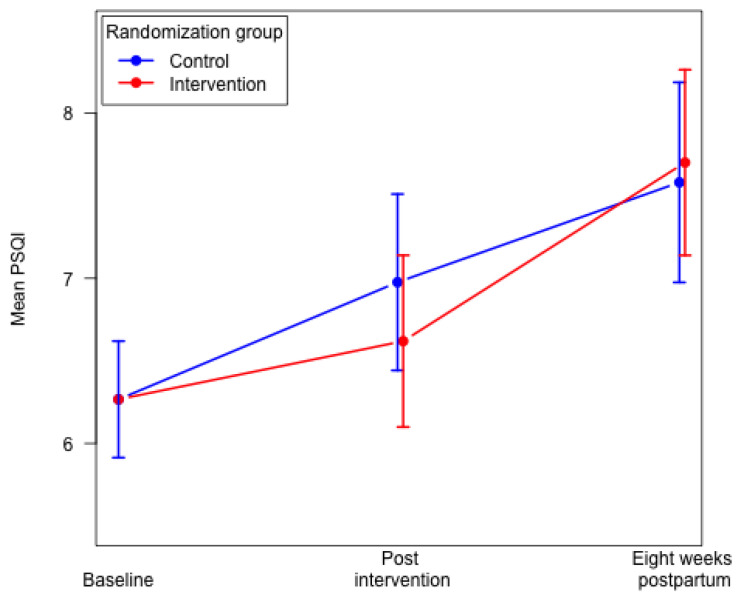
Mean global PSQI score in the intervention group and the control group. Constrained linear mixed model.

**Table 1 ijerph-19-05954-t001:** Baseline characteristics of the study population.

Characteristics	Intervention Group *n* = 143		Control Group *n* = 139	
	Mean	SD	Mean	SD
**Maternal age**, years	31.9	3.8	31.7	3.9
**Body mass index**, kg/m^2^	22.8	3.4	22.7	3.6
**Sleep quality**, PSQI	6.16	2.8	6.37	3.1
**Psychological well-being**, WHO-5	54.4	14.8	56.0	16.4
	** *n* **	**Percentage**	** *N* **	**Percentage**
WHO-5 ≤ 50	55	38.7	42	30.7
WHO-5 > 50	87	61.3	95	69.3
**Nulliparous**	107	74.8	100	71.9
**Living with partner**	137	95.8	132	95.0
**Educational level**				
Advanced degree	72	50.3	74	53.2
3–4 years higher education	47	32.9	40	28.8
1–2 years higher education	9	6.3	5	3.6
Skilled worker	4	2.8	8	5.8
Compulsory education	11	7.7	10	7.2
**Occupation**				
Employed	96	67.1	88	63.3
Unemployed	19	13.3	16	11.5
Student	24	16.8	29	20.9
Other †	4	2.8	6	4.3
**Smoking before pregnancy**	26	18.2	21	15.1
**Smoking in early pregnancy**	2	1.4	0	0
**Physical activity** ≥ 3.5 h a week before pregnancy *	115	80.4	105	75.5
**Chronic disorders** **	26	18	19	14
**History of depression and anxiety**				
Depression within the last 10 years	44	31	39	28
Anxiety within the last 10 years	38	26	42	30
Comorbid depression and anxiety within the last 10 years	61	43	58	42
Antidepressants three months prior to conception and/or during pregnancy	30	21	32	23

† Including stay at home mothers; * The weekly amount of physical activity recommended by The Danish Health Authorities recommendations; ** Chronic disorders: metabolic diseases, respiratory diseases, arthritis, epilepsy and migraine; WHO-5: The five item World Health Organization Well-being Index; PSQI: Pittsburgh sleep quality index; Missing: BMI (2), Educational level (2), WHO-5 (2).

**Table 2 ijerph-19-05954-t002:** PSQI scores at baseline, post-intervention, and postpartum analyzed using Constrained linear mixed model and Logistic regression.

Sleep Parameters	Baseline		Post-Intervention			Eight Weeks pp		
	17–22 wg		29–34 wg					
	IG	CG	IG	CG	*p*	IG	CG	*p*
Global score, mean (CI)	6.16 (5.66–6.66)	6.37 (5.87–6.87)	6.56 (6.05–7.07)	7.00 (6.47–7.53)	0.2	7.69 (7.14–8.25)	7.61 (6.99–8.22)	0.80
Global score > 5, % (n)	57.1 (76)	58.8 (80)	56.8 (71)	64.1 (75)	0.24	78.6 (77)	78.3 (65)	0.97
Component 1 Subjective sleep quality, mean (CI)	1.38 (1.02–1.26)	1.21 (1.09–1.33)	1.2 (1.08–1.34)	1.34 (1.21–1.47)	0.2	1.43 (1.29–1.58)	1.38 (1.21–1.47)	0.50
Subjective sleep quality, very or fairly bad, % (n)	30.8 (41)	31.6 (43)	30.4 (38)	41.9 (49)	0.06	36.7 (36)	41.0 (34)	0.50
	*mean* (*CI*)	*mean* (*CI*)	*mean* (*CI*)	*mean* (*CI*)		*mean* (*CI*)	*mean* (*CI*)	
Component 2 Sleep latency	1.02 (0.87–1.17)	1.13 (0.98–1.28)	0.85 (0.69–1.02)	1.02 (0.86–1.19)	0.28	0.48 (0.33–0.63)	0.63 (0.47–0.78)	0.25
Component 3 Sleep duration	0.64 (0.50–0.77)	0.65 (0.52–0.78)	0.77 (0.63–0.92)	0.95 (0.80–1.10)	0.06	1.58 (1.42–1.75)	1.54 (1.37–1.72)	0.73
Component 4 Habitual sleep efficiency	0.66 (0.51–0.82)	0.76 (0.61–0.91)	0.91 (0.73–1.09)	1.09 (0.90–1.27)	0.26	2.0 (1.81–2.21)	1.70 (1.49–1.92)	0.04
Component 5 Sleep disturbances	1.51 (1.42–1.61)	1.49 (1.40–1.59)	1.62 (1.52–1.72)	1.64 (1.54–1.74)	0.68	1.14 (1.03–1.25)	1.20 (1.09–1.32)	0.38
Component 6 Use of sleeping medication	0.01 (−0.03–0.04)	0.06 (−0.03–0.12)	0.12 (0.04–0.19)	0.04 (−0.04–0.12)	0.14	0.12 (0.02–0.22)	0.10 (−0.01–0.21)	0.69
Component 7 Daytime dysfunction	1.18 (1.06–1.30)	1.07 (0.96–1.18)	1.07 (0.95–1.18)	0.95 (0.83–1.06)	0.29	0.91 (0.78–1.05)	1.05 (0.91–1.20)	0.07

Abbreviations: CG: control group; IG: intervention group; *p*: *p* value; PSQI: Pittsburgh Sleep Quality Index; wg: weeks of gestation; pp: postpartum.

**Table 3 ijerph-19-05954-t003:** Association between subjective sleep quality (PSQI) and psychological well-being (WHO-5) and in the total study population.

		WHO ≤ 50	WHO > 50	*p* Value		WHO ≤ 50	WHO > 50	*p* Value
Baseline		75.2 (65.3–83)	47.3 (39.9–54.8)	<0.0001		7.82 (7.26–8.38)	5.42 (5.02–5.82)	<0.0001
29–34 wg	PSQI > 5	70.3 (59.1–79.4)	55.6 (47.9–63)	0.03	PSQI	7.72 (7.02–8.41)	6.30 (5.82–6.78)	<0.0001
	% (CI)				global score			
					mean (CI)			
8 wk pp		83.7 (71.6–91.3)	75.8 (67.5–82.6)	0.24		8.25 (7.51–9.00)	7.41 (6.90–7.91)	<0.0001

Logistic and linear regression. Abbreviations: PSQI: Pittsburgh Sleep Quality Index; SD: standard deviation; wg: weeks of gestation; WHO-5: World Health Organization Well-Being Index; wk pp: weeks postpartum.

## Data Availability

The data are not publicly available due to limitations in the permission granted from The Danish Data Protection Agency.

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
