# Peer review of "Evaluation of the Effect of Supervised Group Exercise on Self-Reported Sleep Quality in Pregnant Women with or at High Risk of Depression: A Secondary Analysis of a Randomized Controlled Trial"

_ijerph, 2022, doi:10.3390/ijerph19105954_

Round 1
Reviewer 1 Report
Thank you for asking me to review the manuscript, this is an important work. The authors have generally presented the information well.
Some suggestions are provided to improve clarity and make it easier for readers to grasp its content.
Introduction:
The information about recommendation regarding exercise during pregnancy or lack of guidelines is missing. For example, the “Sit Yourself Down”: Women’s Experiences of Negotiating Physical Activity During Pregnancy - Janelle M. Wagnild, Tessa M. Pollard, 2020 (sagepub.com) paper mentioned that the guidelines suggests that pregnant women can have similar amount of exercise as is recommended for non-pregnant women. The authors included some information in section 2.3 as part of description of the control group, but it would be better to come in the introduction already.
Methods
Under the point 2.1, study design, first paragraph, the authors need to provide some information about the objective of the main trial, so it is clear how the objectives are distinct and connect to each other.
Line 17-174: The methods need to have some more information about the advantages of the constrained linear mixed models over the usual liner mixed models. The authors provided a reference, but some information on statistical logic included in the paper would be useful for readers.
Line 152-154: It is not clear why component 1 was dichotomised (whether it was done for descriptive results or some other reason).
The results indicate that sleep quality outcome was analysed as a continuous and binary outcome, which needs to described better in the methods (currently mentioned but can be made more obvious to note). It is also unclear whether apart from overall sleep scores, each of the component was analysed separately (which seems to be the case in table 2).
Results:
All P values across the paper need to use consistent decimal points, reported with at least two decimal points.
It is unclear whether table 2 combined descriptive results and p values from mixed model/ weighted GEE results. Presenting the results for outcomes at various time points separately is informative for readers, but the model results should be presented separately.
In line 188-192, the authors mentioned adjusting for other factors, but did not present the full model results. It would be better to separate out the descriptive and model results, so it is easier for readers to go through the results.
Author Response
Thank you for your constructive and thorough review of our research article. We have carefully revised the manuscript according to the raised comments and suggestions. Please find point-by-point responses to your comments below.
Reviewer 1, Comment 1
Introduction:
The information about recommendation regarding exercise during pregnancy or lack of guidelines is missing. For example, the “Sit Yourself Down”: Women’s Experiences of Negotiating Physical Activity During Pregnancy - Janelle M. Wagnild, Tessa M. Pollard, 2020 (sagepub.com) paper mentioned that the guidelines suggests hat pregnant women can have similar amount of exercise as is recommended for non-pregnant women. The authors included some information in section 2.3 as part of description of the control group, but it would be better to come in the introduction already.
Authors’ response to reviewer 1, comment 1
Thank you for this relevant comment. We agree that it is relevant to describe recommendations regarding exercise during pregnancy in the introduction. We have added the following sentence, lines 70-75:
“It is recommended to conduct at least 30 minutes of moderate-intensity exercise daily through a normal, uncomplicated pregnancy. However, many pregnant women do not meet these recommendations, among other factors due to family obligations and their relative´s perception of exercise being risky during pregnancy. This indicates that women´s physical activity level during pregnancy needs attention.”
Reviewer 1, Comment 2
Methods
Under the point 2.1, study design, first paragraph, the authors need to provide some information about the objective of the main trial, so it is clear how the objectives are distinct and connect to each other.
Authors’ response to reviewer 1, comment 2
Thank you for this comment. We have changed the sentence from:
“The study is a planned secondary analysis of an RCT, the EWE Study, with the primary aim to evaluate the effect of a supervised group exercise intervention on psychological well-being and symptoms of depression among pregnant women with or at high risk of depression.”
To:
“The present study is a planned secondary analysis of an RCT, the EWE Study. The EWE study´s primary objective was to evaluate the effect of a supervised group exercise intervention on psychological well-being and symptoms of depression among pregnant women with or at high risk of depression”.
Reviewer 1, Comment 3
Line 17-174: The methods need to have some more information about the advantages of the constrained linear mixed models over the usual liner mixed models. The authors provided a reference, but some information on statistical logic included in the paper would be useful for readers.
Authors’ response to reviewer 1, comment 3
Thank you for this relevant comment. We agree that describing the advantages of the constrained linear mixed models over the usual liner linear mixed models is important.
We have changed the sentence in section 2.5 Statistical analysis line 196-205from:
“To account for missing responses and to adjust for potential baseline imbalances between the two treatment groups, sum scores were analyzed using constrained linear mixed models considering scores measured at baseline, 29–34 weeks of gestation, and eight weeks postpartum as outcomes”
To:
“To compare the outcomes at 29-34 weeks of gestation and eight weeks postpartum between the two intervention groups, linear mixed models were applied. To gain efficiency in the analyses and to account for potential baseline imbalances, the baseline scores can be included in the analyses. It has been demonstrated that the optimal way to adjust for baseline variables is to include these in the linear mixed models as outcomes rather than as covariates. Due to the randomization, the means of the baseline scores in the two intervention groups are equal by design. Therefore, the linear mixed model also including the baseline scores as outcomes should be constrained to have equal means at baseline in the two groups”
Reviewer 1, Comment 4
Line 152-154: It is not clear why component 1 was dichotomised (whether it was done for descriptive results or some other reason).
Authors’ response to reviewer 1, comment 4
Thank you for this comment. In relation to the dichotomization of item 1, we argue in lines 180-182 that:
“This dichotomization was based on the clinical relevance of discriminating no sleep problems from sleep problems and was conducted similarly in a previous Danish study”.
The main reason for dichotomizing was the clinical relevance, inspired by other research in this area. For this reason, we haven´t added any further to the reason for dichotomization. Please let us know if you disagree with this disposition.
Reviewer 1, Comment 5
The results indicate that sleep quality outcome was analysed as a continuous and binary outcome, which needs to described better in the methods (currently mentioned but can be made more obvious to note). It is also unclear whether apart from overall sleep scores, each of the component was analysed separately (which seems to be the case in table 2).
Authors’ response to reviewer 1, comment 5
Thank you for this comment. In lines 174-177 we have added:
“Beyond the global PSQI score, component 1 also measures subjective sleep quality, here based on the one single question: “During the past month, how would you rate your sleep quality overall?” with the possible answers (very good, fairly good, very bad, or fairly bad)”
For all scores presented in Table 2, estimates are determined from models including the outcomes for all time points (as specified in the table legend).
Reviewer 1, Comment 6
Results:
All P values across the paper need to use consistent decimal points, reported with at least two decimal points.
Authors’ response to reviewer 1, comment 6
Thank you for this comment, the P values are now reported using two decimal points.
Reviewer 1, Comment 7
It is unclear whether table 2 combined descriptive results and p values from mixed model/ weighted GEE results. Presenting the results for outcomes at various time points separately is informative for readers, but the model results should be presented separately.
Authors’ response to reviewer 1, comment 7
Thank you for this comment.
Table 2 includes only model-based results and as such no descriptive statistics are given for the outcomes. This was chosen as the raw means may be biased due to missing data (linear mixed models account for Missing At Random in contrast to the empirical means) and confidence intervals would not be appropriate as these would not account for repeated measures.
We have highlighted this by moving the footnote to the Table 2 heading. It is changed from:
Table 2. - PSQI scores at baseline, post-intervention and postpartum
To:
Table 2. - PSQI scores at baseline, post-intervention and postpartum analyzed using Constrained linear mixed model and Logistic regression
Reviewer 1, Comment 8
In line 188-192, the authors mentioned adjusting for other factors, but did not present the full model results. It would be better to separate out the descriptive and model results, so it is easier for readers to go through the results.
Authors’ response to reviewer 1, comment 8
Thank you for this comment.
None of the analyses were adjusted for other factors than the baseline variables.
Reviewer 2 Report
This is, in summary, an interesting study aimed to evaluate the effect of supervised group physical exercise on self-reported sleep quality in pregnant women with or at high risk of depression, and secondary, to describe the association between sleep quality and psychological well-being during pregnancy and postpartum. The authors found no difference in mean global PSQI score neither at 29-34 weeks, 6.56 in the intervention group and 7.00 in the control group, nor eight weeks postpartum. Furthermore, females with WHO-5 ≤50 reported higher mean global PSQI score at baseline, 7.82, than women with WHO-5 score >50, mean 5.42. Moreover, a significant difference was also present post-intervention and eight weeks postpartum. In addition, no significant effect of group exercise regarding self-reported sleep quality was seen at 29–34 weeks of gestation or postpartum. Finally, low psychological well-being was associated with poor sleep quality during pregnancy and postpartum.
The authors may find my minor comments below.
First, as the authors, throughout the Introduction section, correctly mentioned the importance of inadequate sleep as a prominent and increasing public health problem, they might further stress the possible association between impaired sleep and negative clinical outcomes such as suicidal behavior. Importantly, individuals seeking information and news regarding self-harm and suicidal behaviors are likely to use Internet, particularly when they are affected by impaired sleep. Therefore, given the above information, my suggestion is to include within the manuscript, the study published in 2016 on Psychiatry Res (PMID: 27837725).
Furthermore, as the most relevant aims/objectives underlying the present study have been reported extensively, the main hypotheses might be briefly specified by the authors as well.
In addition, why only 43.6% of pregnant women who were invited to participate provided their written informed consent and completed the baseline questionnaire need to be explained in a detailed manner for the general readership.
Also, the main rationale based on which participants were randomly assigned to supervised group exercise or control group based on a computer-generated random sequence need to be adequately and comprehensively reported.
In addition, the most relevant study limitations/shortcomings should be stressed comprehensively as the main caveats have been only partially reported.
Finally, the authors should identify and focus only on 1-2 most relevant take-home message regarding this paper. Specifically, while the authors reported that supervised group exercise did neither improve self-reported sleep quality at 29–34 weeks of gestation nor at eight weeks postpartum among pregnant women with or at high risk of depression and that overall sleep quality decreased from baseline to eight weeks postpartum and, as expected, that poor sleep was particularly pronounced for the pregnant women with low psychological well-being, they failed, in my opinion, to focus on some conclusive remarks to this specific regard. Specifically, what are the main implications of this manuscript according to these findings? How the present results may be generalized? Here, some additional details/information might be useful for the general readership and should be provided by the authors based on their point of view to this specific regard.
Author Response
Thank you for your constructive and thorough review of our research article. We have carefully revised the manuscript according to the raised comments and suggestions. Please find point-by-point responses to your comments below.
Reviewer 2, Comment 1
First, as the authors, throughout the Introduction section, correctly mentioned the importance of inadequate sleep as a prominent and increasing public health problem, they might further stress the possible association between impaired sleep and negative clinical outcomes such as suicidal behavior.
Importantly, individuals seeking information and news regarding self-harm and suicidal behaviors are likely to use Internet, particularly when they are affected by impaired sleep. Therefore, given the above information, my suggestion is to include within the manuscript, the study published in 2016 on Psychiatry Res (PMID: 27837725).
Authors’ response to reviewer 2, comment 1
Thank you for this comment. We have added the following sentence in the introduction, lines 47-49:
“A recent systematic review stressed the possible association between impaired sleep and self-injury”
However, since we cannot find sufficient evidence for the association between impaired sleep, suicide, and internet searches, we refrain from referring to the suggested article. Please let us know if you disagree with this decision.
Reviewer 2, Comment 2
Furthermore, as the most relevant aims/objectives underlying the present study have been reported extensively, the main hypotheses might be briefly specified by the authors as well.
Authors’ response to reviewer 2, comment 2
Thank you for this comment, we agree that it is relevant to specify the main hypotheses, especially the second part concerning the association between poor mental health and poor sleep quality. It is described as bidirectional, where inadequate sleep can be both a causal contributor to and a symptom of disorders such as depression and anxiety. Unfortunately, with our study design, it was not possible to become wiser about the direction of the association.
We have added the sentence:
“Secondly, we hypothesized that the strength of the sleep-psychological health association would be moderate”.
Reviewer 2, Comment 3
In addition, why only 43.6% of pregnant women who were invited to participate provided their written informed consent and completed the baseline questionnaire need to be explained in a detailed manner for the general readership.
Authors’ response to reviewer 2, comment 3
Thank you, we appreciate this very relevant comment
“The main reason for declining to participate was lack of time (Figure 1)”.
“A large proportion of eligible women declined to participate, and unfortunately, the ethics committee of the capital region of Denmark did not permit us to collect baseline characteristics of the invited women choosing not to participate.”
and
“Time factors are known reasons for not participating in a research project which might explain why more multiparous than nulliparous women did not accept the invitation.”
Reviewer 2, Comment 4
Also, the main rationale based on which participants were randomly assigned to supervised group exercise or control group based on a computer-generated random sequence need to be adequately and comprehensively reported.
Authors’ response to reviewer 2, comment 4
Thank you for this relevant comment. We have changed the sentence from;
“The participants were randomly assigned to either supervised group exercise (n=143) or control group (n=139) (Figure 1) by a computer-generated random sequence using permuted block sizes (four, six, or eight)”.
To
‘To provide a fair comparison between the intervention and the control group the distribution of known and unknown prognostic factors was balanced, on average, at baseline by using randomly permuted block randomization (block size four, six, or eight), ensuring proper allocation sequence concealment. The participants were randomly assigned to either supervised group exercise (n=143) or control group (n=139) (Figure 1).”
Reviewer 2, Comment 5
In addition, the most relevant study limitations/shortcomings should be stressed comprehensively as the main caveats have been only partially reported.
Authors’ response to reviewer 2, comment 5
Thank you, we agree.
We have changed the sentence from:
“The study population was well-educated, had normal BMI, was largely physically active before, during, and after pregnancy, and was proficient in Danish, limiting the generalizability of the results to other populations”.
To:
“The study population was well-educated, had normal BMI, was largely physically active before, during, and after pregnancy, was primarily primiparous, and was proficient in Danish, limiting the generalizability of the results to other populations”.
As well as changing the sentence in lines 406-408.:
“The PSQI is a validated patient-reported outcome measure widely used as an outcome in the obstetric field”.
To:
“The PSQI is a validated patient-reported outcome measure widely used as an outcome in the obstetric field, however, during pregnancy, the related comorbid conditions of being pregnant are likely to influence the results”.
Further, we have added the following sentence in the strength and limitation section:
“As the control group reported the same weekly amount of physical activity as the intervention group, a Hawthorne effect cannot be ruled out. Participation in the study itself might have provided motivation for some of the women to increase their exercise level, however, to reduce this risk we did not introduce the control group to the exercise program”
Reviewer 2, Comment 6
Finally, the authors should identify and focus only on 1-2 most relevant take-home message regarding this paper. Specifically, while the authors reported that supervised group exercise did neither improve self-reported sleep quality at 29–34 weeks of gestation nor at eight weeks postpartum among pregnant women with or at high risk of depression and that overall sleep quality decreased from baseline to eight weeks postpartum and, as expected, that poor sleep was particularly pronounced for the pregnant women with low psychological well-being, they failed, in my opinion, to focus on some conclusive remarks to this specific regard. Specifically, what are the main implications of this manuscript according to these findings? How the present results may be generalized? Here, some additional details/information might be useful for the general readership and should be provided by the authors based on their point of view to this specific regard.
Authors’ response to reviewer 2, comment 7
Thank you for this very important comment.
In lines 434-440 we have added the following:
“Sleep complaints are common during pregnancy, but it needs attention in antenatal care that sleep quality further decreases postpartum and that poor sleep quality is particularly pronounced for women with depression or low psychological well-being. While poor sleep is both a symptom of and a causal contributor to depression, health care professionals should give attention to the risk of a vicious circle. Further studies are needed to examine the possible effect of exercise on sleep quality in other populations of pregnant women
Reviewer 3 Report
Int. J. Environ. Res. Public Health 2022
Evaluation of the effect of supervised group exercise on self-reported sleep quality in pregnant women with or at high risk of depression. A secondary analysis of a randomized controlled trial.
General evaluation
This paper addressed a very important topic: Association between sleep quality and psychological well-beeing.
The aim of this study was to evaluate the effect of supervised group physical exercise on self-reported sleep quality in pregnant women with or at high risk of depression, and secondary, to describe the association between sleep quality and psychological well-being during pregnancy and postpartum.
This study is important and has relevance for all pregnant women in the populations, for heath provider and for researchers. Following elements can be considered to strengthen the paper further:
Abstract and Keywords:
OK
Background:
Nice introduction and background. Please add a theoretical construct and the hypotheses. What are the hypothesized direction of the association? How strong should the association be, what is reported in the literature?
Spelling: Line 45 Underscore
Spelling: Line 46 Underscore
Methods:
Please add some more details about the information of the women. For example was there a Psychoeducation session to understand the intervention. This is relevant for CBI and I assume this is the same for sleep quality improvements interventions.
Results:
OK
Discussion
Are these results a surprise? I miss this in the background. Is there no literature about this topic?
Pleas add more about confounders. This is very relevant, as exercise is a good intervention.
Line 320: Maternity leave 10 months. In this 10 Months women experience a hormonal chaos in their bodies – exercises are too week to interfere with hormones.
Line 338: The PSQI is a general tool and does not include all the body, hormonal and feeling changes pre and post-partum. There are multiple factors and a cutoff change would not solve the problem.
Please add in the discussion that sleep quality decreases during the gestational age due to weight change, Body change, different sleep position, nocturia, restless Leg syndrome, cramps… There is a huge amount of confounding factors.
Line 262: Actigraphy would not strengthen the study. It is just impossible with this sample size to do actigraphy.
Line 369: It is not about the validation. The PSQI is valid in pregnant women. All the related comorbid condition of being pregnant are confounding all your results.
Author Response
Thank you for your constructive and thorough review of our research article. We have carefully revised the manuscript according to the raised comments and suggestions. Please find point-by-point responses to your comments below.
Abstract and Keywords:
OK
Background:
Reviewer 3, Comment 1
Nice introduction and background. Please add a theoretical construct and the hypotheses. What are the hypothesized direction of the association? How strong should the association be, what is reported in the literature?
Please add a theoretical construct and the hypotheses
Authors’ response to reviewer 3, comment 1
“Poor sleep quality has been associated with an increased risk of cesarean section [10], prolonged labor [11], and perinatal depression”.
To:
“Poor sleep quality has been associated with an increased risk of cesarean section [10], prolonged labor [11], and a recent systematic review found a moderate association between poor sleep and perinatal depression”.
“Among individuals with severe mental illness, a systematic review from 2019 found that exercise had a strong, positive effect on sleep quality.
The basic mechanism underlying the exercise-sleep relationship is not fully understood, but it has been suggested that the gradual decline in body temperature occurring after exercising contributes to drowsiness and facilitates sleep. Another possible mechanism is that exercise improves sleep quality by reducing anxiety”.
And in line 76-78 changed the sentence:
“Specifically, in pregnancy,
“A study found that
And in lines 79-83 changed the sentence from:
“A randomized controlled trial (RCT) (n=92) found that yoga reduced sleep disturbances in pregnant women with depression [21], and larger RCTs but replications are required to determine the clinical potential of exercise as a means to improve sleep quality among pregnant women with or at high risk of depression”.
To:
“A randomized controlled trial (RCT) (n=92) found a small effect of yoga in relation to sleep disturbances in pregnant women with depression, and larger RCTs and replications are required to determine the clinical potential of exercise as a means to improve sleep quality among pregnant women with or at high risk of depression”.
In relation to our hypothesis we added the following sentence in lines 86-87:
“Secondly, we hypothesized that the strengths of the sleep-psychological health association would be moderate ”
Reviewer 3, Comment 2
Spelling: Line 45 Underscore
Spelling: Line 46 Underscore
Authors’ response to reviewer 3, comment 2
This has now been corrected
Reviewer 3, Comment 3
Methods:
Please add some more details about the information of the women. For example was there a Psychoeducation session to understand the intervention. This is relevant for CBI and I assume this is the same for sleep quality improvements interventions.
Authors’ response to reviewer 3, comment 3
, Indeed it is relevant to describe which information the women got. The women were informed about the aim but not about the potential mechanism between exercise and mental health/sleep. There was not a psychoeducation session to understand the intervention, the information was focused on how to perform the exercise in the desired intensity.
In lines 151-153 we have added the sentence:
“The women in the intervention group were carefully introduced to the meaning of 60-70% of one repetition maximum (RM) by the physiotherapists and individual suitable weight for exercises was found for each participant”.
Results:
OK
Reviewer 3, Comment 4
Discussion
Are these results a surprise? I miss this in the background. Is there no literature about this topic?
Authors’ response to reviewer 3, comment 4
Thank you for this important comment. We agree that it was missing in the background. Please see our answer to comment 1.
Further, we changed the sentence from:
“We found that women with reduced psychological well-being (WHO-5 score ≤50) at baseline reported a significantly higher mean global PSQI score than women with high psychological well-being (WHO-5 score >50), which is in line with previous studies finding an association between low psychological well-being and poor sleep”
To:
“As expected, we found that women with reduced psychological well-being (WHO-5 score ≤50) at baseline reported a significantly higher mean global PSQI score than women with high psychological well-being (WHO-5 score >50). This is in line with previous studies finding an association between low psychological well-being and poor sleep, however, we found a larger proportion of women and a stronger association than expected”
Reviewer 3, Comment 5
Pleas add more about confounders. This is very relevant, as exercise is a good intervention.
Line 320: Maternity leave 10 months. In this 10 Months women experience a hormonal chaos in their bodies – exercises are too week to interfere with hormones.
Authors’ response to reviewer 3, comment 5
In line 354-363 we have changed the sentence from:
“Mothers in Denmark are on average on maternity leave 10 months after giving birth, so we have no reason to believe that returning to work have influenced our result”
To:
“Mothers in Denmark are on average on maternity leave ten months after giving birth, so we have no reason to believe that returning to work has influenced our results. It cannot be ruled out that mechanical, hormonal, vascular, and metabolic changes occurring during pregnancy and postpartum have influenced the results. However, due to the randomization we assume these factors to be equally distributed in both groups”.
Reviewer 3, Comment 6
Thank you for this relevant comment.
Line 338: The PSQI is a general tool and does not include all the body, hormonal and feeling changes pre and post-partum. There are multiple factors and a cutoff change would not solve the problem.
Authors’ response to reviewer 3, comment 6
Thank you, we appreciate this relevant point.
In lines 372-376 we have changed the sentence from:
“However, our and other studies, findings of an average PSQI score above the cut-off score used to differentiate good and poor sleepers, may indicate that a cut-off score of five is inappropriate in a population of pregnant women”
To:
“However, in the present and other studies, findings of an average PSQI score above the cut-off score used to differentiate good and poor sleepers, may reflect that the PSQI is a generic tool that does not take all the e.g. hormonal and metabolic changes that occur during pregnancy and postpartum, into account.”
Reviewer 3, Comment 7
Please add in the discussion that sleep quality decreases during the gestational age due to weight change, Body change, different sleep position, nocturia, restless Leg syndrome, cramps… There is a huge amount of confounding factors.
Authors’ response to reviewer 3, comment 7
Thank you.
We have changed the sentence from:
“This finding is in line with a recent study measuring sleep quality using PSQI and with a meta-analysis”
To:
“This finding is in line with a recent study measuring sleep quality using PSQI [38] and with a meta-analysis [8] and as described in a review factors such as weight change, nocturia, and restless leg syndrome are often seen during pregnancy and can cause poor sleep quality”.
Reviewer 3, Comment 8
Line 262: Actigraphy would not strengthen the study. It is just impossible with this sample size to do actigraphy.
Authors’ response to reviewer 3, comment 8
Thank you, we agree with this point of view.
We changed the sentence from:
“An objective measurement of sleep using actigraphy could have strengthened the study”
To:
“However, using an objective measurement of sleep as actigraphy would be difficult with the present sample size”.
Reviewer 3, Comment 9
Line 369: It is not about the validation. The PSQI is valid in pregnant women. All the related comorbid conditions of being pregnant are confounding all your results.
Authors’ response to reviewer 3, comment 9
Thank you for this relevant comment.
We changed the sentence from:
“The PSQI is a validated patient-reported outcome measure widely used as an outcome in the obstetric field [28]. However, it would have strengthened the study if the PSQI had been validated in a population of pregnant women, also to elucidate if a higher cut off score, differentiating good and poor sleepers, would be relevant”
To:
The PSQI is a validated patient-reported outcome measure widely used as an outcome in the obstetric field [28], however, during pregnancy, the related comorbid conditions of being pregnant are likely to bias the results.
Round 2
Reviewer 2 Report
In the revised paper, the authors addressed sufficiently most of the major questions raised by Reviewers.